# Navigating Food Fraud: A Survey of Nigerian Consumer Knowledge and Attitudes

**DOI:** 10.3390/foods13203270

**Published:** 2024-10-15

**Authors:** Helen Onyeaka, Amarachukwu Anyogu, Olumide A. Odeyemi, Michael Ukwuru Ukwuru, Ukpai Eze, Folayemi J. Isaac-Bamgboye, Christian K. Anumudu, Oluwabunmi O. Akinwunmi, Olufemi Peter Sotayo, Yemisi A. Jeff-Agboola

**Affiliations:** 1School of Chemical Engineering, University of Birmingham, Birmingham B15 2TT, UK; anumuduchristian@gmail.com; 2School of Biomedical Sciences, University of West London, London W5 5RF, UK; amara.anyogu@gmail.com; 3Office of Research Services, Research Division, University of Tasmania, Launceston, TAS 7248, Australia; oluodeyemi@gmail.com; 4Department of Food Science and Technology, The Federal Polytechnic Idah, Idah 27110, Nigeria; mikeukwuru@gmail.com; 5Chester Medical School, University of Chester, Chester CH2 1BR, UK; 6Department of Food Science and Technology, Federal University of Technology Akure, Akure 340110, Nigeria; yemistyle@gmail.com; 7Department of Food Science and Technology, Faculty of Agriculture, Bowen University, Iwo 232102, Nigeria; bunmiakinwunmi2019@gmail.com; 8Department of Microbiology, Faculty of Science, University of Lagos, Akoka-Yaba, Lagos 101017, Nigeria; olufemisotayo@gmail.com; 9Department of Microbiology, University of Medical Sciences, Ondo City 351103, Nigeria; yjeffagboola@yahoo.com

**Keywords:** economically motivated adulteration, food fraud, food safety, Nigeria, consumer perception

## Abstract

Food fraud presents a growing threat globally, impacting consumer health; food safety; and brand credibility. The key motivation for food fraud is usually an economic or financial advantage. Economically motivated food fraud (EMA) is the illegal deception, deliberate and intentional substitution or addition of a substance in a food product, which may pose a possible health risk to unsuspecting consumers. This study aims to assess the awareness and attitudes of Nigerian consumers towards food authenticity and the regulatory measures in place to combat food adulteration. The study comprised 2160 respondents who were asked about their knowledge of high-profile food fraud incidents, frequently adulterated products, and their opinions of the effects of food fraud and preventative measures. The findings of this study revealed that more than a third of respondents were unfamiliar with the term “food fraud”. However, they were aware of local high-profile cases, such as “plastic rice” and counterfeit dairy products. Most participants relied on point-of-sale information for assessing food authenticity, with street vendors being viewed as potential sources of food adulteration. The study also found that more than half of the respondents were concerned about food fraud regardless of whether it involved local or imported products. A wide variety of food items were identified as prone to adulteration or fraudulent activity. These included alcoholic drinks, dairy products, honey, rice, and tomato paste, amongst others. These findings highlight the need for improved consumer awareness, regulatory effectiveness, and remediation efforts to combat food fraud in Nigeria.

## 1. Introduction

Food fraud is a global issue that poses a threat to public health, economic growth, and sustainable development [1,2,3,4]. According to [3], “Food fraud is a collective term used to encompass the deliberate and intentional substitution, addition, tampering or misrepresentation of food, food ingredients or food packaging: of false or misleading statements made about a product, for economic gain”. Food supply chains have become increasingly complex due to the need for rapid, global distribution systems for raw materials, ingredients, and products. However, this complexity contributes to the growing scale of food fraud and its impact on consumers [3,5,6].

High-profile incidents of food fraud include the adulteration of infant formula with melamine in China and the substitution of horsemeat in beef products, which affected several European countries [7,8]. In addition, frequent reports of fraudulent activities in fish and fish products, milk and milk products, and oil and honey products worldwide have exposed flaws in global food regulatory [9,10,11].

Nigeria is not immune to food fraud. As a result of the increasing incidents of the adulteration of food and food products in Nigeria, the National Agency for Food and Drug Administration and Control (NAFDAC) enacted the Recall, Disposal and Handling of Unwholesome and Adulterated food and Food Products Regulations in 2019 [12]. In this regulation, reference [12] defined adulterated food and food products as “food that has been prepared, packed, or held under condition that is unfit for human consumption whereby it may have become contaminated with filth, chemical or microbial substances and it may have been rendered injurious to health. It includes the addition or subtraction to or from food so that the natural composition and quality of the food is affected”. Fraudulent acts usually involve changing the composition of a food product by dilution or substitution with a lower-quality product or adulterating it with unapproved ingredients. Food smuggling and diversion, mislabeling, revalidation and repackaging of expired food products, and counterfeiting have also been reported [13,14]. Although there is a scarcity of scientific documentation of food fraud, media reports have noted that fraudulent activities occur frequently in Nigerian food supply chains. In 2016, the Nigeria Customs Service reported that it had confiscated 2.5 tons of ‘plastic rice’ imported into the country [15]. In the same year, a study conducted by NAFDAC, reported that 91% of imported tomato paste sampled fell below the 28% minimum of tomato content required [16]. More recently, NAFDAC also alerted the public on the distribution and sale of unregistered and expired energy drinks [17]. Other food products implicated in food fraud include spices, honey, and alcoholic drinks [14,18,19,20]. These incidents have raised concerns about the safety and authenticity of food products in the Nigerian market.

A relatively recent example of food fraud is that of Nestlé, which was convicted of employing illegal methods of filtering for its mineral water products. The company has been said to be using unapproved filtration methods for many years and this has led to the concerns about their product integrity and quality. Research in France shows that since the 1990s, Nestle has treated polluted water using processes banned in food processing and has been marketing it as “Natural Mineral Water” without disclosure [21]. This has elicited a lot of outrage from the public and demand for much more strict measures from food producers. It is hoped that the case can act as a reminder of just how important proper scrutiny is in preventing risk to the consumer and a weakening of confidence in food products.

Fraudulent food products can pose severe health risks, as consumers may unknowingly consume substandard or unsafe ingredients. Mislabeling of fish products, for example, may lead to potential allergen exposure or consumption of fish with higher levels of contaminants [3]. The consumption of adulterated alcoholic beverages can result in adverse health effects including death [22,23]. Furthermore, inaccuracies in the nutritional content of food products can contribute to malnutrition. Long-term illnesses can lead to an increased burden on already fragile healthcare systems.

Food fraud also has significant economic implications, including financial losses for consumers, businesses, and the overall economy. Food fraud undermines public trust in the food supply chain. Consumers may seek alternative products to those associated with the affected supplier or manufacturer or question the motives of those tasked with consumer protection [24]. Consumer confidence in meat authenticity was impacted after the horsemeat scandal [25]. It is estimated that food fraud costs the global food industry $30–40 billion annually [3]. While specific figures for Nigeria may be lacking, the economic impact of food fraud is likely to be substantial.

The vulnerability of food supply chains to fraud is influenced by the ease of opportunities to commit fraud, economic drivers, and the effectiveness of control measures for preventing fraud [26]. During price shifts due to regional or global supply shortages, consumers with limited purchasing power are more likely to choose illegal or substandard food products because these are cheaper [27,28]. In Nigeria, like many African countries, there is poor infrastructure for the monitoring and enforcement of food standards, where these exist, by policymakers and regulatory agencies. In addition, the technology for identifying fake or adulterated food is often absent [29,30]. This can create an environment that allows more opportunities for fraud to occur and remain undetected.

Food choices are made every day, and consumers often make these decisions without complete knowledge of a product’s quality and safety [31]. Consequently, consumers are the main victims of food fraud, yet there are scant studies on consumer awareness and attitudes towards food fraud, particularly in low-income settings [32]. This insight is crucial for developing effective strategies to combat food fraud and protect public health. The findings of this study will contribute to the existing literature on food fraud and consumer behavior, specifically in the Nigerian context. This knowledge can help policymakers and stakeholders in the food industry to develop effective strategies to prevent and mitigate food fraud in Africa’s largest economy. Additionally, it can inform consumers on how to identify and avoid food fraud, thereby reducing the health and economic risks associated with the consumption of fraudulent food products. As food fraud is a global issue, research findings from Nigeria could inform similar studies in other countries facing similar challenges. The aim of this study was thus to investigate consumer awareness and attitudes towards food fraud, and, in particular, perceptions of the efficacy of the government and regulator in tackling food fraud in Nigeria.

## 2. Methods

Ethical approval for the study was obtained from the Ethics and Research Committee of Abia State University Teaching Hospital, Abia, Nigeria (ABSUTH/MAC/117/VOL II/79). Participants gave informed consent via the statement, “I am aware that my responses are confidential, and I agree to participate in this survey”, where an affirmative reply was required to enter the survey. The participants were able to withdraw from the survey at any time without giving a reason.

### 2.1. Survey Questionnaire and Pilot Testing

The study involved two phases. A clear data collection protocol was followed, and all stages of the study were documented. This included decisions about sampling, data collection methods, and analysis procedures, which ensured transparency. The findings were reported transparently, with a focus on both statistically significant results and the potential limitations of the study (such as the absence of effect-size measures).

In phase one, the questionnaire was developed following the modification of previous studies, including a systematic review [31,33,34,35] and consultations among the researchers as the subject matter experts. The draft questionnaire consisted of three sections with twenty-five questions as follows: (1) socio-demographic—six questions, (2) food fraud awareness—eight questions and (3) food fraud attitude—eleven questions. The draft questionnaire was piloted among 44 participants who were not included in the final study. The purpose of the pilot study was to improve, standardize and ensure the questionnaire is easy to understand and adaptable to local context. The questionnaire was distributed via convenient sampling. The participants were asked to read the questionnaire and to provide their thoughts on the time taken to complete the questionnaire, clarity of the instructions for completing the questionnaire, clarity and mutual exclusiveness of the answers to the questions, and suggestions on clarification of the instructions and how to improve the format of the questions. Based on the feedback provided, the sociodemographic section was retained, while the section on food fraud awareness was modified. An additional option was provided to question five, and a qualitative question to provide examples of food fraud that the participants were aware of was added. This, therefore, increased the total number of questions in this section to nine from eight. In the third section of the questionnaire, questions 1 to 6 and 8–10 were modified for clarity and inclusion of mislabeled food, while questions 7 and 11 were not modified. The final questionnaire contained twenty-six questions structured into three parts:
i.Socio-demographic characteristics including age, gender, marital status, educational level, income, and geopolitical zone.ii.Food fraud awareness, including knowledge of high-profile international and domestic cases of food fraud and relevant sources of information about food fraud.iii.Food fraud attitudes aimed at sampling the opinion of the respondents on the effects of food fraud, preventive measures, and rationale for food adulteration.

### 2.2. Sampling and Distribution of the Questionnaire

In the second phase of the study, the final questionnaire was disseminated from 28 July 2021 to 28 February 2022. A dual dissemination approach was adopted to maximize the number of respondents reached for the survey using both online (WhatsApp and email) [36] and in-person methods. In acknowledging the potential bias in our survey, we recognize that the initial sample size might not have been fully representative due to the online dissemination method, which may have skewed the responses towards a more educated or internet-accessible demographic. However, we addressed this by expanding the survey to cover a wider geographic area. Additionally, we supplemented the distribution with face-to-face engagements, ensuring a more diverse and inclusive sample, which enhanced the reliability of our findings. A multistage sampling approach, combining stratified sampling (to ensure representation across Nigeria’s six geopolitical zones) and convenience sampling (to capture accessible populations), was employed. This sampling method helped ensure that the sample was both diverse and representative of different regions and demographic groups. The questionnaire was designed based on existing validated tools, ensuring that the questions were relevant and clear. This minimized the risk of misunderstanding or leading questions that could bias the results. For in-person surveys, survey administrators were trained to maintain a neutral tone and avoid influencing participants’ responses. This helped ensure that the data collected reflected genuine consumer attitudes rather than being influenced by the survey process.

Participants were fully informed about the study’s purpose through the distribution of consent forms before participation. This ensured that respondents provided data voluntarily and with a full understanding of the research objectives, enhancing the authenticity of the responses. For the in-person survey, a printed copy of the questionnaire was distributed in markets and educational institutions across the six (6) geopolitical zones of Nigeria. Using both online (WhatsApp, email) and in-person distribution methods ensured that data were collected from a wide variety of respondents, including those who might not have access to online platforms. This approach minimized biases that might arise from relying solely on one method, ensuring comprehensive coverage across different segments of the population. To maintain authenticity, the data collection process was standardized across all regions. For instance, the same version of the questionnaire was used for both online and in-person responses, and survey administrators were trained to handle in-person surveys consistently. By comparing data collected through these different channels, inconsistencies were identified and addressed, which strengthened the credibility of the findings.

## 3. Statistical Analysis

The data obtained from this survey were analyzed using Statistical Package for the Social Sciences (IBM SPSS version 28). Descriptive statistics expressed in percentages were used to analyzed the social-demographic information provided by the participants. Analysis of Variance (ANOVA: *p* = 0.05) was further used to describe the effect of the socio-demographic information on the dependent variables. The analysis was conducted using descriptive statistics and ANOVA, which are robust tools for comparing group differences and identifying patterns in the data. These techniques are widely accepted for analyzing survey data, and their use ensured that the findings are statistically sound. Before analysis, the data were cleaned to remove any incomplete or invalid responses, ensuring that the dataset was accurate and reliable. Any inconsistencies or outliers were addressed systematically to avoid skewing the results. All statistical analyses were conducted using established software (SPSS) (IBM SPSS version 28), and detailed records were kept regarding the analysis process, including the assumptions made and any adjustments applied. This ensured that the results are reproducible and can be independently verified.

## 4. Results

### 4.1. Sociodemographic Characteristics of Respondents

A total of 2160 respondents correctly completed the questionnaire, of which 56.9% were female and 43% male (Figure 1a). Age-wise, the largest group of respondents was the younger demographic group (18–34 years), at 70%. Less than 18% were between the ages of 35–44 years, 6.9% were between the ages of 45–54, and less than 3% were above 55 years (Figure 1b). Most of the respondents had achieved some level of education. About 72% of respondents declared achieving a graduate degree or higher, while more than 27% said they had a secondary school certificate. Less than 1% of respondents reported no formal education or ending their education at primary school (Figure 1c).

To survey the location of respondents, Nigeria was separated into six geo-political zones (Figure 1d). Most respondents (35.1%) were from the South-Eastern region, followed by the South-West (25.9%) and South-South (16.9%) regions. The lowest participation was in the Northern regions with 9.4%, 9.1% and 3.6% from the Northeast, North Central and Northwest regions respectively. The survey asked respondents to declare their monthly household income. Almost 28% reported having a monthly household income of less than N30,000 ($65), which represents the minimum wage in Nigeria. About 34% of respondents have a monthly household income between N30,001 and N100,000 ($216), while almost 33% declared an income of N100,001 to N250,000 ($540). Only 5.9% of the respondents reported a household income above N500,000 ($1090) (Figure 1e). Most respondents were single (63%) compared to married (35%) (Figure 1f).

### 4.2. Consumer Awareness of Food Fraud

Table 1 presents the results of consumers’ awareness of the concept of food fraud and food fraud-related incidents. Of our sample (n = 2160), about a third of respondents were not aware of the term ‘food fraud.’ When probed, respondents showed varying levels of awareness of specific incidents of food fraud. They were more aware of local high-profile incidents of food fraud such as ‘plastic rice’ (81.9%) compared to global incidents, e.g., melamine (33%) and horsemeat (44.4%) scandals. Regarding fraudulent activities involving specific food products, the highest level of awareness was noted for counterfeit dairy products (43.3%) and dilution fraud in cooking oils (34.5%). Only a quarter of respondents were aware of adulteration or counterfeiting of alcoholic drinks, substitution fraud in spices (21%), adulteration of honey (16%), and locally produced rice re-bagged as imported brands (14%).

Respondents were also asked about how they found out about any fraudulent food products. Most respondents obtained this information from the point of purchase (39.3%), followed by the manufacturer (19.1%), social media or news outlets (13.1%), or communication from regulatory agencies (10.1%). Only 7% of respondents reported that they would find out this information from their research or investigation. Street vendors were the most selected option (68.3%), followed by open markets (45.8%) when asked to identify where fraudulent activities are likely to occur in the supply chain. Only 5.4% and 6.5% of respondents reported that these activities occurred in manufacturing facilities or warehouses, respectively. When asked about the recalls and safety alerts system from the National Agency for Food and Drug Administration and Control (NAFDAC), almost half of the respondents (49.2%) did not know about it.

The survey generated interesting results when correlating socioeconomic determinants with awareness of food fraud. Education, household income, and geo-political zone had the most significant (*p* < 0.05) influence on awareness of the concept of food fraud (Table 2). Awareness of the melamine and horsemeat scandals was significantly correlated with age, gender, education, and marital status (*p* < 0.02). However, for the local incident of ‘plastic’ rice, gender, education, and geo-political zone (*p* < 0.002) were the strongest indicators for awareness (Table 2). Table 2 further presents an analysis of the level of awareness of food fraud and recall incidences among respondents concerning various demographic variables.

The analysis revealed that age did not significantly impact awareness and recall of food fraud incidents. The F-statistic of 0.846 with a *p*-value of 0.469 obtained suggests that there is no statistically significant difference in awareness and recall among respondents of different age groups. Similarly, gender was found to have no significant effect on food fraud awareness and recall, with an F-statistic of 1.428 and a *p*-value of 0.233, indicating that there is no statistically significant difference in awareness and recall between genders. The level of education also did not appear to be a significant factor influencing food fraud awareness and recall among respondents, with an F-statistic of 1.189 and a *p*-value of 0.313. In contrast, marital status was found to have a statistically significant influence on food fraud awareness and recall. The F-statistic of 0.608 with a *p*-value of 0.032 indicates that marital status did affect awareness and recall. The geopolitical zone of respondents also did not significantly impact food fraud awareness and recall. The F-statistic of 1.331 with a *p*-value of 0.263 suggests that there are no statistically significant differences in awareness and recall among respondents from different geopolitical zones. Household income was found to have a marginal impact on food fraud awareness and recall. The F-statistic of 1.688 with a *p*-value of 0.167 indicates that there may be slight differences in awareness and recall based on household income, but these differences were not highly significant.

### 4.3. Consumer Attitudes towards Food Fraud

Of the 2160 respondents, the majority (85.6%) expressed concern about fraudulent food products, with 59.1% strongly agreeing and 26.5% agreeing. Similar concerns were noted for both locally produced (85%) and imported (79%) food products (Table 3). Regarding responsibility for protecting consumers from food fraud, 64.1% of respondents believed that regulatory agencies like NAFDAC and the Standards Organisation of Nigeria (SON) were the most competent, while 55.6% believed that food producers, manufacturers, and sellers should take the lead. Interestingly, almost 70% of respondents believed it was up to consumers to protect themselves when shopping or eating at a restaurant. Although the majority (68.4%) of respondents agreed that food fraud can cause serious health challenges, including death, almost 20% of respondents disagreed with this (Table 3).

Consumer attitudes towards the role of the government and its agencies in tackling food fraud were also explored. About 74% of respondents did not agree or strongly disagreed that the government was doing enough to create awareness about food fraud. Despite the existence of NAFDAC numbers, which are meant to evidence the authenticity of a product, almost half of the respondents (49.3%) disagreed with or were neutral to this statement. Similarly, while 33.7% of respondents believed that it was easy to report suspected cases of food fraud to regulatory agencies, over 60% either disagreed or were neutral. An interesting perspective on consumer attitude towards food fraud was highlighted in the response to the question; “During challenging economic times, it is acceptable for food sellers to sell counterfeit or adulterated food items as long as these foods do not cause harm”. The survey results show a complex consumer perspective on food fraud during economic hardship. While a majority (52%) strongly disagree, a combined 21.1% (9.8% + 11.3%) of respondents believe and agree with the statement (Table 3). Pearson correlation showed that the only significant (*p* = 0.012) association between attitudes to food fraud and demographic variables related to household income (Table 4). Interestingly, this was a negative correlation, suggesting that higher-income earners expressed the most negative attitudes. There was no significant correlation between any of the socio-demographic characteristics and food fraud awareness (Table 4). The survey respondents provided examples of food products they believe are commonly adulterated. Their responses revealed a wide variety of foods subject to adulteration. To visualize these data, a word cloud (Figure 2) was generated, where the size of each food item indicates the frequency of its mention. Foods that were more commonly cited appear larger, while those mentioned less frequently are smaller.

The two major classifications of the examples of adulterated foods mentioned by the respondents are shown in Figure 3. The breakdown of food fraud incidents in this study indicated that 44% of foods involved were processed foods and 56% raw foods, which has several potential implications and significance. Figure 4 illustrates the findings regarding the nature of food fraud incidents reported by the respondents. It reveals a striking pattern, with 2% of respondents mentioning mislabeling as an issue and a substantial 98% reporting cases of adulteration. This stark contrast in percentages suggests that adulteration is a predominant concern among consumers, significantly overshadowing mislabeling in terms of reported food fraud incidents.

Figure 5 provides valuable insights into the distribution of food types affected by fraud, as reported by respondents. The data reveals that rice is the most mentioned food product, accounting for 22% of the reported cases. Palm oil follows closely at 20%, and honey at 10%. Several other food products including drinks, dairy meat, poultry, and seafood were also identified by respondents.

## 5. Discussion

This research explored consumer awareness and attitudes towards food fraud in Nigeria. To ensure that each zone was adequately represented, stratified sampling was ideal. By dividing the population into these strata (geopolitical zones), we systematically ensured geographic diversity, capturing different cultural, economic, and social contexts that might affect consumer attitudes towards food fraud. This approach ensured that the sample was not biased towards one region or population group, which was crucial given the diverse socio-economic conditions across the zones. Stratification helped achieve representativeness of the Nigerian population, making the findings more generalizable.

Within each geopolitical zone, convenience sampling was appropriate for reaching participants both online and in-person. Convenience sampling involved selecting respondents based on ease of access—such as through WhatsApp, email, and in-person distribution at markets and educational institutions. This method helped maximize the number of respondents by taking advantage of accessible populations, especially given the logistical challenges of reaching every individual within a large, geographically dispersed population. By using a dual dissemination approach (online and in-person), the study effectively captured a broad range of respondents, enhancing the sample size while balancing practical constraints.

Similar studies conducted in different countries have noted that consumer awareness and attitudes towards food fraud can be influenced by social, cultural, and economic characteristics [36]. Limited consumer awareness of food fraud has been highlighted as a critical challenge limiting the detection and prevention of food fraud in Africa [30]. In this study, more than a third of the participants were unfamiliar with the term ‘food fraud.’ In addition, there was generally poor awareness of issues related to some commonly adulterated foods, as observed in the responses in Table 1. This supports the observations made by [29] when investigating consumer knowledge of food adulteration in Ghana. Lower awareness of food fraud in certain regions can be attributed to various socio-economic, educational, cultural, and infrastructural factors. Educational disparities play a significant role, as regions with limited access to formal education and specific food safety training are less informed about the risks of food fraud. Economic hardship also affects awareness, with consumers in poorer areas often prioritizing affordability over safety, making them more tolerant of adulteration. Cultural norms may normalize food adulteration, especially in informal markets, reducing vigilance. Additionally, some regions have weaker regulatory oversight, where food safety laws are poorly enforced, allowing fraud to proliferate without consumers being aware. In such areas, informal markets may dominate, offering fewer quality controls. Regions with limited infrastructure and access to information also have lower awareness, as consumers may lack access to media or communication channels that disseminate information on food safety. A lack of government prioritization of food safety can also contribute, with few public campaigns or insufficient funding to address food fraud issues. Moreover, consumer perception plays a role: in regions where food fraud is not seen as an immediate health risk, people are less likely to be concerned about it. Sometimes, they may even view adulteration as a necessary trade-off for cheaper food. Globalization affects exposure to food fraud, as more isolated regions might not be aware of international scandals or the complexities of global supply chains that drive awareness in more developed regions. To improve food fraud awareness, targeted education, stronger regulation, and better access to information are crucial.

Socio-demographic determinants may influence consumer awareness and, thus, vulnerability to fraudulent and counterfeit food products [35]. Therefore, this study investigated if awareness and attitudes toward food fraud differed according to demographic factors. Nigeria has a population of 229 million people who are predominantly young, with a median age of 17.2 years, of which the majority (54%) are classified as urban dwellers [37]. Most of the participants in this study were under 35 years old (Figure 1). These data reflect the country’s demographic profile, as it has been proposed that approximately 70% of the population is under 30 [38]. This observation may suggest that this section of the population has more interest or concern about food fraud compared to other groups. However, it should be noted that online questionnaires may skew towards younger adults who are more likely to have Internet access and know-how [39]. In terms of education, in this study, almost all respondents (99%) had at least a secondary school education. This corroborates the observations of [40], who reported that most of the respondents from several African countries were educated at the tertiary level. Our results are comparable to reports from Canada [35], Israel, and Germany [36]. However, these countries rank in the Very High Human Development category in the UN’s Human Development Index, while Nigeria scores in the Low Human Development category [41]. This index combines measurements of education, life expectancy, and per capita income. It is difficult to find accurate demographic data from Nigeria from official sources. In 2018, the World Bank reported the literacy rate in Nigeria was 62% [42]. Similar to the age demographics, the use of an online questionnaire presented only in English may have selected for highly educated participants. In a similar study in Ghana, authors observed that 63% of their respondents had completed secondary school [29]. However, the questionnaire was administered in person, and questions were translated for respondents who could not read or understand.

Comparisons of socio-demographic determinants yielded some surprising results on food fraud awareness. For example, previous studies have noted that women tend to be more interested in health-related matters linked to food, e.g., nutrition and food safety [43]. Women also tend to act as primary household shoppers, and therefore, it is expected that they would have higher awareness of fraudulent activities in food supply chains. In this study, no significant differences in awareness of some frequently adulterated and counterfeited products were observed between men and women. However, marital status yielded significant results for most of the questions asked. The reasons for these observations are unclear and will need to be probed further. Education, household income and age were the most significant determinants of awareness of local and international high-profile food fraud cases. This could be because these demographics have better access to information and increased knowledge about food systems in general. From the data obtained in this study, overall demographic factors have a limited impact on food fraud awareness and recall among Nigerian consumers. This implies that food fraud awareness campaigns and interventions should be designed to reach a broad audience without focusing excessively on specific demographic characteristics. It is important to note that while statistical significance may be present in some cases, the practical significance (effect size) is often small, suggesting that these differences may not have a substantial real-world impact.

While demographic factors may have a limited impact on food fraud awareness and recall, the findings underscore the importance of comprehensive awareness campaigns, tailored communication strategies, improvements in recall systems, promotion of consumer empowerment, and multi-stakeholder collaboration in addressing food fraud risks, enhancing consumer protection, and advancing food safety and security in Nigeria. Further qualitative research may be needed to understand the nuances of how different demographic groups access information about food fraud.

Interestingly, most participants noted that their main source of information related to food fraud was from the point of purchase or manufacturer and not from regulatory agents or news outlets. At the same time, they identified retailers such as street vendors and market sellers as the point in the supply chain where mislabeling, counterfeiting, and adulteration were most likely to occur. Sellers may have a vested interest or be actively involved in fraudulent activity; therefore, regulatory organizations should provide independent, accurate information about the authenticity of food to consumers. Only 51% indicated knowledge of the national recall alert system, suggesting that campaigning and broadcasting related to food fraud incidents is low.

Furthermore, results suggest street vendors are a crucial source of information regarding counterfeit or adulterated food products. Many street food vendors in Nigeria may not have received formal education or training in food safety and hygiene practices. This can result in a lack of awareness regarding the importance of safe food handling, storage, and preparation. Therefore, there is a need for targeted education and awareness campaigns for this population.

The low level of awareness of the recalls and safety alerts system from NAFDAC among the respondents in the current study is concerning. NAFDAC may not be effectively utilizing various communication channels to disseminate recalls and safety alerts to the public. There may be a lack of visibility of NAFDAC’s messages through traditional media, online platforms, community networks, and other channels that reach diverse demographic groups across Nigeria. Improving the reach and accessibility of NAFDAC’s communications can enhance awareness and responsiveness to safety alerts. NAFDAC may need to reassess its information dissemination strategies to ensure that safety alerts and recall notices are communicated, promptly, and comprehensively to consumers, retailers, healthcare professionals, and other stakeholders in the food supply chain. This may involve adopting multi-modal approaches, including text messages, social media posts, radio broadcasts, print media, and community outreach programs to reach a wider audience. NAFDAC may need to enhance transparency and accountability in its recall and safety alert processes to build trust and confidence among stakeholders. This includes providing timely updates, explanations, and follow-up actions on safety concerns, investigations, and enforcement actions related to recalled products. Open communication channels and stakeholder engagement mechanisms can facilitate dialog, feedback, and collaborative problem-solving.

The findings of this study, together with previous studies [44,45] highlight the urgent need for improved food safety and security in Nigeria. The high levels of food fraud and low levels of awareness of food safety regulations and enforcement mechanisms among consumers and street food vendors in Nigeria suggest that there is a pressing need for education, regulation, and enforcement efforts to protect consumers from the health risks associated with adulterated or counterfeit food products. The capacity and effectiveness of regulatory agencies such as NAFDAC and the Standards Organization of Nigeria (SON) should be enhanced to enforce existing food safety regulations and standards. This includes developing and implementing robust inspection, monitoring, and surveillance systems, as well as imposing deterrent penalties for non-compliance with food safety regulations. Providing training, technical assistance, and incentives to food producers, processors, distributors, and street food vendors to adopt and adhere to internationally recognized good manufacturing practices and food hygiene standards is required. This can help mitigate contamination risks, improve product quality, and enhance consumer confidence in the safety of food products. Implementation of systems for enhanced food traceability, product labeling, and supply chain transparency to track the origin, processing, and distribution of food products from farm to fork. Leveraging technologies such as blockchain, QR codes, and electronic monitoring systems can facilitate real-time monitoring, authentication, and traceability of food products, enabling rapid response to food safety incidents and enhancing consumer trust.

In this study, most respondents were concerned about fraudulent food products, regardless of whether they were produced domestically or imported. These findings support previous studies that have shown that consumers are generally concerned about food fraud irrespective of the region of origin [3]. Consumers are increasingly aware of the potential health risks associated with consuming adulterated, counterfeit, or contaminated food products. Incidents of foodborne illnesses, outbreaks, and recalls linked to fraudulent practices have heightened public awareness and underscored the importance of vigilance and caution when making food choices. A large proportion of the respondents believe that government agencies such as NAFDAC and SON are the most competent to protect Nigerians from counterfeit, adulterated or mislabeled foods. This is expected, as NAFDAC is responsible for regulating and controlling the manufacture, importation, exportation, distribution, advertisement, and sale of food, drugs, cosmetics, medical devices, and chemicals in Nigeria. SON, on the other hand, is responsible for developing and promoting industrial and product standards. Both agencies have specific mandates and expertise related to food safety, quality, and standards as well as the legal authority and regulatory power to set and enforce standards for food safety and quality in Nigeria. They can create and implement regulations, conduct inspections, and enforce compliance with these standards. Both agencies have laboratories and testing facilities where they can analyze food products for authenticity, quality, and safety, issue certifications and product approvals based on rigorous testing and inspection processes. These agencies also conduct regular market surveillance activities to monitor the quality of products on the market. They can identify and are empowered by law to act against counterfeit, adulterated, or mislabeled foods and products.

Interestingly, almost half of the respondents believe it is up to consumers to protect themselves from counterfeit, adulterated, or mislabeled foods when shopping or eating at a restaurant. Advocates of this viewpoint often argue that consumers have the freedom to make informed choices when purchasing or consuming food products. They believe that consumers should take personal responsibility for their choices and should be aware of the potential risks associated with food consumption. Some proponents of this perspective emphasize the role of market forces and competition in ensuring the quality and safety of food products. They argue that businesses are incentivized to provide safe and high-quality products to attract and retain customers.

The implication of 48% of respondents in the Nigerian survey strongly agreeing with the statement “I believe it is up to consumers to protect themselves from counterfeit, adulterated, or mislabeled foods” is twofold: It suggests a significant portion of Nigerians believe the onus of safe food lies with them, rather than solely on regulatory bodies or food producers. This could be due to a lack of trust in existing food safety measures or a perception that consumers need to be extra vigilant; While consumer vigilance is important, placing the entire burden on them can be problematic. Consumers may not have the expertise to identify counterfeit, adulterated, or mislabeled foods. Also, access to reliable information about food safety practices and identification methods might be limited, especially in rural areas. Thus, food producers and regulatory bodies still have the primary responsibility to ensure food safety and prevent fraud. While consumer awareness is important, the primary responsibility for ensuring safe food lies with regulatory bodies who need to enforce stricter regulations and improve enforcement mechanisms to deter food fraud and food producers that must uphold ethical practices and prioritize food safety throughout the production chain. Educating consumers on how to identify safe food choices can empower them to make informed decisions but not grant them the responsibility to tackle fraud. The viewpoint expressed by respondents here reflects a perspective rooted in individual responsibility, consumer empowerment, and market dynamics. Advocates of this viewpoint emphasize the importance of consumer education and awareness in enabling individuals to make informed choices about the food products they purchase and consume. By educating consumers about food safety risks, labeling requirements, and detection methods, they can recognize and avoid potentially hazardous or fraudulent products. This knowledge and information about food fraud, safety and quality can lead to more informed choices. Consumer education campaigns can help individuals recognize warning signs, such as unusual packaging or labeling, and make safer decisions. However, it is important to note that expecting consumers to solely bear the burden of protecting themselves from food fraud may have limitations and challenges. Consumers may face barriers such as limited access to information, lack of resources, cognitive biases, and misinformation that impede their ability to make fully informed choices.

The study found that a significant proportion of the respondents (49.1%) strongly agree that food fraud can cause serious health challenges, including death. This finding is consistent with previous studies that have shown that food fraud can have serious health implications [3]. The perspective expressed by the respondents reflects a heightened awareness and recognition of the potential risks and consequences associated with fraudulent food practices. The respondents’ acknowledgement of the serious health risks posed by food fraud underscores the gravity of the issue and the importance of prioritizing food safety measures. It is a call for action that underscores the urgency and importance of addressing the root causes of fraudulent practices, strengthening regulatory oversight, and enhancing consumer protection measures. It highlights the need for collaborative efforts among government agencies, industry stakeholders, civil society organizations, and consumers to promote transparency and accountability in the food supply chain.

The study found that a combined 21.1% of the respondents agree/strongly agree that during challenging economic times, it is acceptable for food sellers to sell counterfeit or adulterated food items if these foods do not cause harm. Although 52.3% disagree, the proportion of respondents that agree with the statement indicates that a significant portion of Nigerians might be willing to tolerate food fraud under economic pressure. The finding that some consumers are willing to accept food adulteration during economic hardships, provided it does not cause harm, raises important questions about consumer priorities and perceptions of food safety. During times of financial strain, consumers may prioritize affordability over the integrity of food products. They may view minor adulteration as a necessary compromise to meet their basic needs, as long as it does not seem to present an immediate health risk. This pragmatic approach reveals how economic insecurity can influence consumer tolerance for practices that would otherwise be unacceptable. Some consumers might assume that food adulteration, when it occurs, is still regulated in a way that minimizes harmful risks. They may believe that if a product is still on the shelves, it must meet a minimum safety standard. This assumption can reduce their concern over adulteration, especially when they do not perceive any direct or immediate consequences. In some societies, minor food adulteration might be normalized, especially if it is seen as a way to stretch limited resources. The cultural context and past experiences of consumers with food safety could shape their attitudes towards what is deemed acceptable or harmful.

These findings highlight a potential justification for food fraud by sellers during economic hardship. This can threaten consumer health, even if the adulteration is not immediately harmful. The survey suggests that economic hardship can pressure food vendors into unsafe practices. This emphasizes the need for social safety nets and economic support programs to alleviate financial strain and reduce the incentive for food fraud. Food insecurity remains a challenge in many African countries and food availability may be perceived as more important than food authenticity. This finding highlights deep-seated ethical, moral, and public health concerns within the food industry. Accepting the sale of counterfeit or adulterated food items, even if they do not cause immediate harm, represents a violation of consumers’ rights to safe, authentic, and accurately labeled food products. Allowing the sale of counterfeit or adulterated food items perpetuates unethical and dishonest practices within the food supply chain. It prioritizes short-term economic gains over long-term public health, undermines consumer trust, and erodes the integrity of the food industry. Ethical standards dictate that businesses and food sellers have a moral obligation to uphold honesty, transparency, and accountability in their dealings with consumers. While some respondents may perceive counterfeit or adulterated foods that do not cause immediate harm as acceptable during challenging economic times, the long-term health risks and consequences of consuming such products cannot be overlooked. It reflects broader ethical dilemmas and societal values regarding the trade-offs between economic survival and ethical integrity. It underscores the need for ethical reflection, dialog, and collective action to reconcile competing interests, uphold moral principles, and prioritize the well-being of consumers and society.

The prevalence of food fraud in raw and processed foods raises concerns about the potential health risks associated with the consumption of adulterated products. Food fraud in raw ingredients can expose vulnerabilities in the supply chain. It may highlight weaknesses in the sourcing, transportation, and handling of these ingredients. This finding underscores the vulnerability of the entire food supply chain to fraudulent practices. Consumers may unknowingly purchase and consume adulterated, contaminated, or mislabeled food products, posing serious risks to their health and well-being. The presence of harmful substances, allergens, toxins, pathogens, and other contaminants in fraudulent food products can lead to foodborne illnesses, allergic reactions, and other adverse health effects. The finding exposes the complexity and interconnectedness of the food supply chain, where fraudulent activities can occur at various stages of production, processing, distribution, and retailing. Raw ingredients and agricultural commodities may be susceptible to adulteration, while processed foods may be vulnerable to mislabeling, substitution, or dilution with inferior or unsafe ingredients. The widespread nature of food fraud underscores the need for comprehensive strategies to mitigate risks and ensure the integrity and safety of the food supply chain. Addressing food fraud in raw and processed foods poses regulatory challenges for government agencies tasked with overseeing food safety and enforcement. Regulatory frameworks may need to be strengthened, expanded, or adapted to effectively detect, prevent, and deter fraudulent practices across diverse food categories and product types. Enhancing regulatory capacity, surveillance capabilities, and collaboration with industry stakeholders are essential for safeguarding consumer health and ensuring compliance with food safety standards.

Previous studies have also highlighted the issue of food fraud in Nigeria. A study by (Opia, 2020) found that food fraud was a significant problem in Nigeria, with adulteration and substitution of food products being common practices. This finding aligns with global tendencies in which food fraud poses critical risks to public health, customer acceptance as true and economic integrity inside the food delivery chain. In a study on how worried consumers are about food fraud [40], it was revealed that Ghanaian and Nigerian consumers tend to score higher on the measure of food fraud concern suggesting that they were less confident in the safety and quality of the food they consume. These studies, together with the current study, emphasize the need for concerted efforts to improve food safety and security in the country. This research underscores the urgent need for concerted efforts to improve food protection and safety in Nigeria. Addressing food fraud calls for complete strategies that contain collaboration among government groups, enterprise stakeholders, civil society agencies, academia, and clients to reinforce regulatory frameworks, enhance surveillance and enforcement mechanisms, and promote transparency and accountability at some point in the food supply chain.

Identifying and addressing these vulnerabilities is essential to safeguard the integrity of the entire food system. Recognizing the distribution of food fraud can guide targeted interventions. Authorities, industry stakeholders, and consumer advocacy groups can develop strategies to tackle the specific challenges associated with raw and processed foods. For example, they may implement stricter testing protocols for raw ingredients or enhance transparency in the processing of foods. This is paramount because ensuring the integrity of both types of foods is vital for consumer protection.

The overwhelming prevalence of adulteration and reported food fraud incidents is a matter of grave concern. The data obtained from this present study underscore the need for stringent measures to combat adulteration and protect public health. The high percentage of respondents reporting adulteration highlights the potential erosion of consumer confidence in the authenticity and safety of the food they purchase. Rebuilding and maintaining consumer trust is essential for the food industry. While mislabeling represents a smaller percentage, it should not be dismissed. It may indicate a need for improved consumer education and awareness regarding how to identify and report mislabeling incidents. Importantly, the prominence of rice and palm oil as the most frequently reported foods in fraud incidents is of major concern as these foods are staples in many diets, and their adulteration or fraudulent practices can have far-reaching consequences, including adverse health effects and economic losses for consumers. Furthermore, it can affect a wide range of products as they are utilized as ingredients in a vast array of dishes. Thus, there is a need for comprehensive surveillance and regulatory measures across the food industry.

The effect of food fraud affects not only consumers but even honest companies operating in the sector become affected as well. In the same way, fraudulent firms depress their costs and have higher sales while triggering harm to other genuine competitors. As a result, the ethical players are edged out of the market, and, in the most severe of circumstances, may be forced to shut down business. The desire to stay relevant has been implicated, forcing honest organization to engage in similar illegal conduct, thereby creating a culture of fraud in that particular niche. Moreover, after the fraud is unveiled, there are severe consequences that an organization receives, as well as the overall market. Consumers may even carry forward this issue to a point where they begin to distrust the authenticity of the product category all together if the guilty party is found. The negative implications for an industry targeted by a misleading competitor may mean that consumers switch to the use of close substitutes, thereby harming only the ethical competitors. The impact on consumer trust is potentially very damaging, possibly permanently reducing both demand and sales rates. Therefore, food fraud is not only a negative for the consumer; it is also dangerous for any ethical business model and could lead to the collapse of large industries.

The distribution of food fraud incidents between processed and raw foods highlights the broad scope of the issue. It indicates that food fraud is not limited to a specific category of products, but affects both raw ingredients and processed items. This suggests that the problem is widespread across various stages of the food supply chain. The higher percentage of raw foods being affected by food fraud indicates a significant concern. When raw ingredients are adulterated, it can lead to safety issues as these ingredients are used in various processed foods. Ensuring the authenticity and safety of raw ingredients is crucial to prevent contamination of processed products. The prevalence of food fraud in both raw and processed foods can have a substantial economic impact. Adulteration or fraud can lead to financial losses for both producers and consumers. In processed foods, it can result in higher production costs and lost revenue, while in raw foods, it can affect agricultural or primary production sectors. The distribution of food fraud across both categories underscores the need for robust regulatory and quality control measures. Authorities must monitor and enforce food safety standards not only at the processing stage, but also throughout the entire supply chain, including the sourcing of raw ingredients. The safety and authenticity of food products are critical for public health and consumer confidence. While this study has produced several statistically significant results, it is important to note that the analysis does not currently include measures of effect size. Despite the absence of effect size calculations, the statistically significant results of this study provide a strong foundation for practical policy recommendations. By focusing on the real-world implications of consumer attitudes toward food adulteration, policymakers can begin to address the underlying issues that drive acceptance of adulteration, particularly during economic hardship and across various demographic groups.

## 6. Conclusions

In conclusion, this study has shed light on the pervasive nature of food fraud in Nigeria, revealing concerning trends and implications for consumer health, safety, and trust within the food supply chain. The findings underscore the prevalence of food fraud across various demographic groups and food categories, highlighting the complexity and multifaceted nature of the issue.

It is important to note that most respondents were young and highly educated, which introduces selection bias in the study. Therefore, extrapolating our results to all Nigerian consumers should be undertaken with caution. However, our findings reveal insightful information about the nature of food fraud in Nigeria and the importance of tackling this problem. Key findings include the high levels of consumer awareness and concern about food fraud, the recognition of serious health risks associated with fraudulent food products, and the prevalence of food fraud in raw and processed foods. Furthermore, the study identified demographic factors such as age, gender, education, income, marital status, and geographic location that may influence awareness, perceptions, and attitudes towards food fraud among Nigerian consumers.

The implications of these findings are far-reaching, emphasizing the urgent need for comprehensive strategies to combat food fraud, strengthen regulatory oversight, and promote consumer protection in Nigeria. Addressing the root causes of food fraud requires collaborative efforts among government agencies, industry stakeholders, civil society organizations, and consumers to enhance transparency, integrity, and accountability throughout the food supply chain.

Despite the valuable insights gained from this study, several gaps in knowledge remain, warranting further investigation and research. Future studies could explore the effectiveness of regulatory frameworks and enforcement mechanisms in detecting, preventing, and deterring food fraud in Nigeria; the socio-economic drivers and incentives underlying food fraud practices among food producers, manufacturers, distributors, and retailers; the impact of food fraud on public health outcomes, including the prevalence of foodborne illnesses, hospitalizations, and mortality rates; the role of consumer education, awareness campaigns, and risk communication strategies in empowering consumers to make informed food choices and detect fraudulent practices; the adoption of technological innovations, traceability systems, and supply chain management solutions to enhance transparency and authenticity in the food industry; the socio-cultural, political, and economic factors shaping perceptions of food fraud and attitudes towards regulatory interventions among Nigerian consumers.

By addressing these knowledge gaps and advancing evidence-based research, policymakers, industry stakeholders, and researchers can develop targeted interventions, regulatory reforms, and best practices to strengthen food safety, protect consumer interests, and promote public health in Nigeria and beyond. Future studies must prioritize collaboration, interdisciplinary approaches, and community engagement to build a more resilient, transparent, and sustainable food system that prioritizes the well-being of consumers and the integrity of the food supply chain.

## Figures and Tables

**Figure 1 foods-13-03270-f001:**
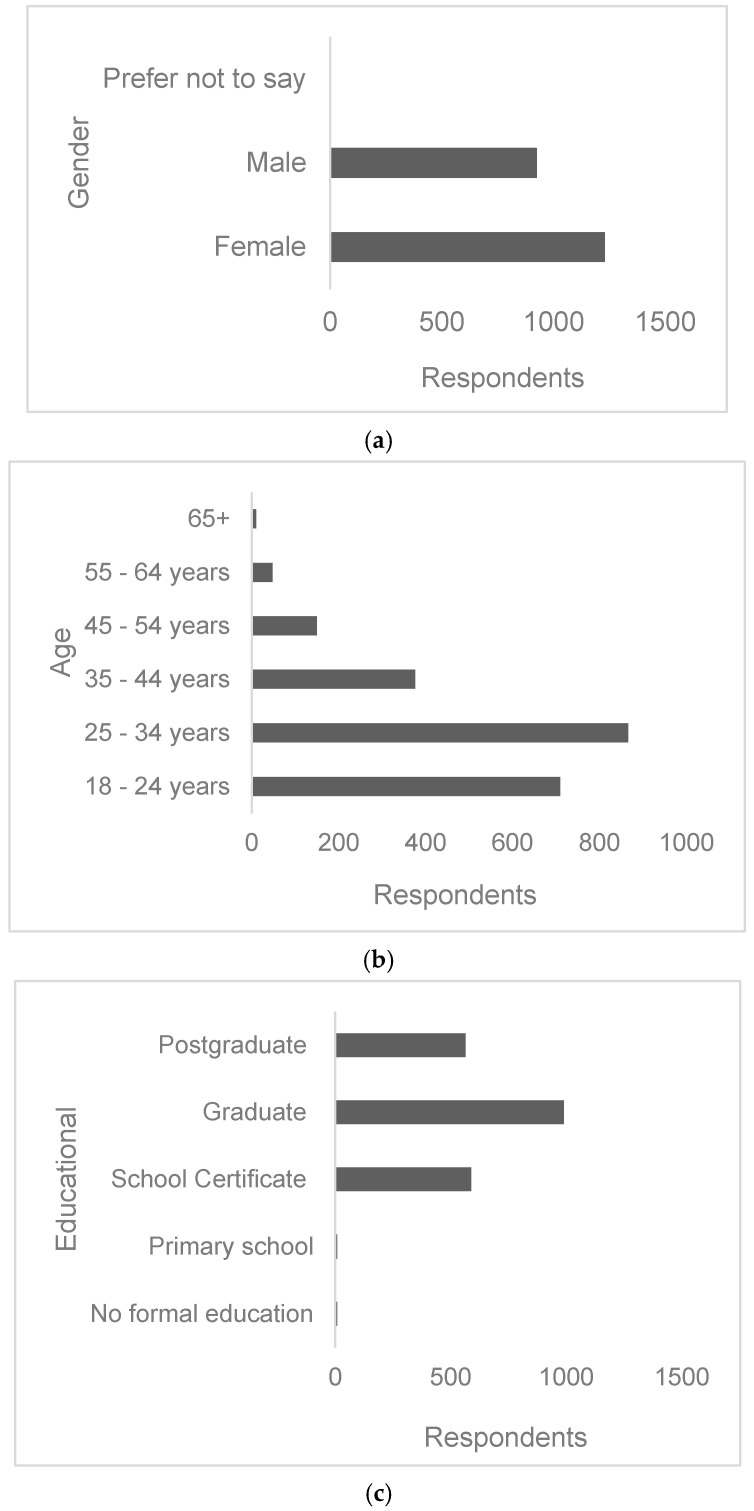
(**a**–**f**) Socio-demographic characteristics of respondents (n = 2160).

**Figure 2 foods-13-03270-f002:**
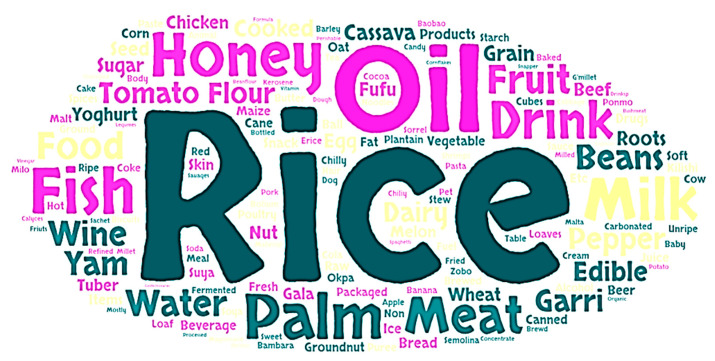
Word cloud of examples of adulterated food provided by the respondent. The size of the food indicated frequency of mention by the respondent.

**Figure 3 foods-13-03270-f003:**
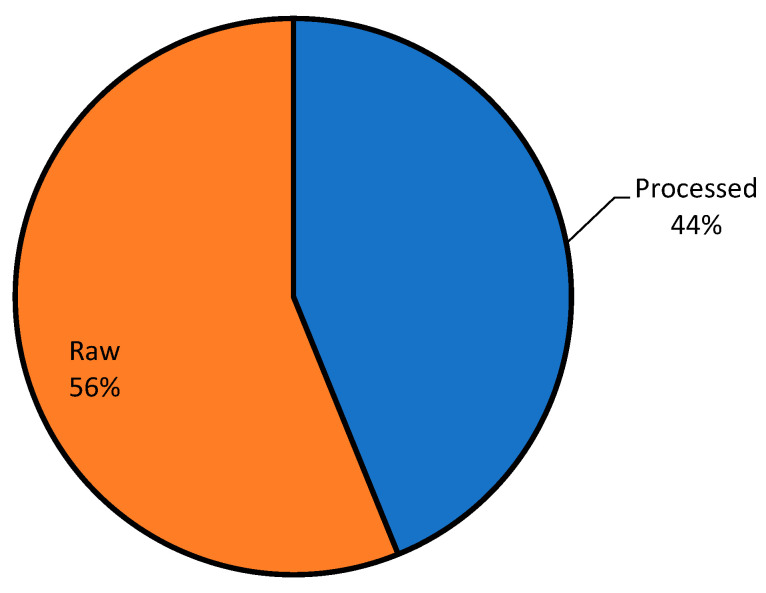
Condition of food investigated.

**Figure 4 foods-13-03270-f004:**
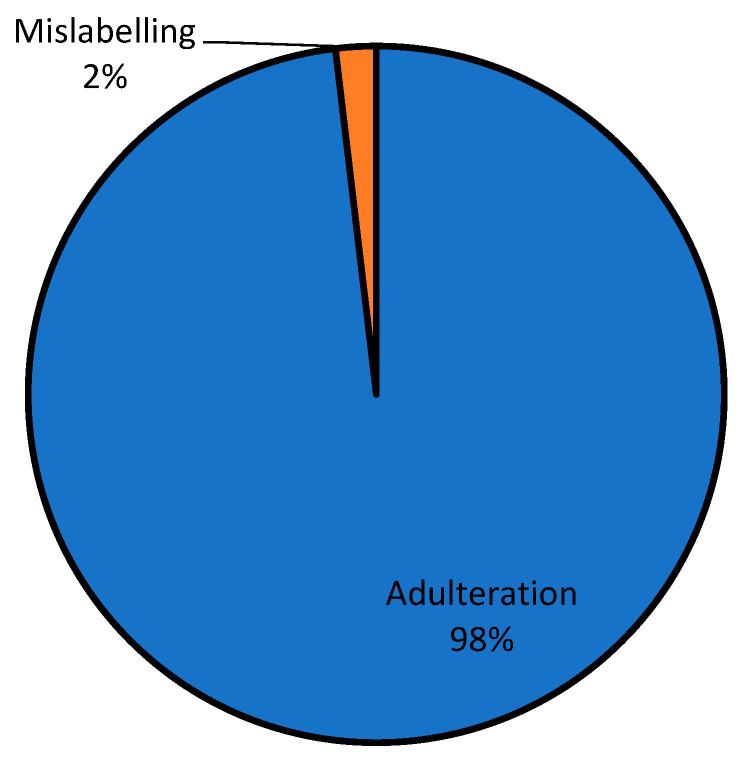
Type of food fraud.

**Figure 5 foods-13-03270-f005:**
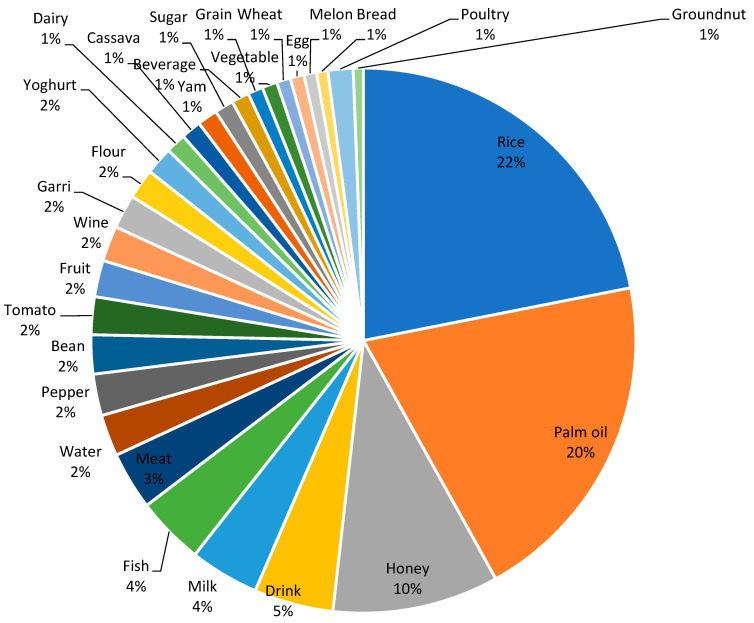
% of Foods in fraud.

**Table 1 foods-13-03270-t001:** Awareness of global and local food fraud incidents and recalls.

Question	% (n)	
	Yes	No
Before reading the information provided about this study, were you aware of the term ‘food fraud’?	64.1 (1384)	35.9 (776)
Are you aware of the milk adulteration with melamine scandal?	33 (712)	67 (1448)
Are you aware of the sale of ‘plastic rice’?	81.9 (1768)	18.1 (392)
Are you aware of the horsemeat scandal?	44.4 (959)	55.6 (1201)
**Which of the following practices are you aware of?**		
a. Locally produced rice rebagged as imported rice brands.	13.9 (301)	77.5 (1674)
b. Milk and yoghurt repackaged as popular brands.	43.3 (936)	50.5 (1090)
c. Adulteration or counterfeiting of cooking oils, e.g., adding colour dye to palm oil.	34.5 (746)	59.3 (1280)
d. Adulteration or counterfeiting of alcoholic drinks.	25.2 (544)	68.6 (1482)
e. Adulteration of honey.	15.5 (335)	78.3 (1691)
f. Addition of ground red kola nut or dyes to dry ground red pepper.	21.1 (455)	72.7 (1571)
**How likely are you to find out about a counterfeit or adulterated food product? Please select as many as apply.**		
Through information from the place where the food was purchased.	39.3 (848)	54.5 (1178)
Through information from the manufacturer (company or business) producing the food.	19.1 (413)	74.7 (1613)
Through social media or news outlets.	13.1 (283)	80.6 (1741)
Through communication from a regulatory agency, e.g., NAFDAC or Standards Organisation of Nigeria (SON)	10.1 (218)	83.7 (1807)
Through your research or investigation.	6.9 (149)	86.9 (1876)
**Where do you think those food products are likely to be adulterated/counterfeited/mislabeled, etc.? Please select as many as apply**		
Street vendors	68.3 (1475)	25.5 (551)
Open market	45.8 (990)	48 (1036)
Supermarket or store	12.8 (277)	81 (1749)
Restaurant	8 (172)	85.8 (1854)
Warehouse	6.5 (140)	87.3 (1886)
Manufacturing facility	5.4 (117)	88.4 (1909)
Are you aware of the recalls and safety alerts system from NAFDAC?	50.8 (1098)	49.2 (1062)

**Table 2 foods-13-03270-t002:** Awareness of food fraud and recall among respondents.

**(a). Awareness of food fraud and recall among respondents.**												
**Variable**	**Statistic**		**Before Reading the Information Provided about this Study, Were You Aware of the Term ‘Food Fraud’**		**Are You Aware of the Milk Adulteration with Melamine Scandal?**		**Are You Aware of the Sale of ‘Plastic Rice’?**		**Are You Aware of the Horsemeat Scandal?**		**Are You Aware of the Recalls & Safety Alerts System from NAFDAC?**	
**Age**												
	Sum of Squares		3.09		3.121		0.434		6.961		0.996	
	df		5		5		5		5		5	
	Mean Square		0.618		0.624		0.087		1.392		0.199	
	F		2.694		2.836		0.584		5.698		0.797	
	Sig.		0.02 *		0.015 *		0.713		<0.001 *		0.552	
**Gender**												
	Sum of Squares		0.098		0.133		2.405		1.87		1.385	
	df		2		2		2		2		2	
	Mean Square		0.049		0.066		1.203		0.935		0.693	
	F		0.212		0.3		8.146		3.796		2.774	
	Sig.		0.809		0.741		<0.001 *		0.023 *		0.063	
**Education**												
	Sum of Squares		6.774		6.774		4.217		7.001		0.54	
	df		4		4		4		4		4	
	Mean Square		1.694		1.694		1.054		1.75		0.135	
	F		7.441		7.757		7.174		7.168		0.54	
	Sig.		<0.001 *		<0.001 *		<0.001 *		<0.001 *		0.707	
**Marital status**												
	Sum of Squares		0.231		0.786		1.03		7.354		0.876	
	df		4		4		4		4		4	
	Mean Square		0.058		0.197		0.258		1.839		0.219	
	F		0.251		0.889		1.735		7.534		0.876	
	Sig.		0.909		0.47		0.139		<0.001 *		0.477	
**Geo-political zone**												
	Sum of Squares		2.631		0.774		2.811		1.074		1.345	
	df		5		5		5		5		5	
	Mean Square		0.526		0.155		0.562		0.215		0.269	
	F		2.292		0.699		3.808		0.87		1.076	
	Sig.		0.043 *		0.624		0.002 *		0.501		0.372	
**Household income**												
	Sum of Squares		9.92		1.689		1.783		3.749		1.216	
	df		4		4		4		4		4	
	Mean Square		2.48		0.422		0.446		0.937		0.304	
	F		10.967		1.913		3.011		3.815		1.216	
	Sig.		<0.001 *		0.106		0.017 *		0.004 *		0.302	
**(b). Awareness of food fraud and recall among respondents.**												
**Variables**	**ANOVA**		**Local Rice Rebagged as Imported Rice.**		**Milk and Yoghurt Repackaged as Popular Brands.**		**Adulteration or Counterfeiting of Cooking Oils, e.g., Adding Colour Dye to Palm Oil.**		**Adulteration or Counterfeiting of Alcoholic Drinks.**		**Adulteration of Honey.**	**Addition of Ground Red Kola Nut or Dyes to Dry Ground Red Pepper.**
**Age**												
	Sum of Squares		2.727		4.55		3.361		1.157		1.379	2.49
	df		3		1		1		1		1	1
	Mean Square		0.909		4.55		3.361		1.157		1.379	2.49
	F		0.846		4.242		3.131		1.077		1.284	2.319
	Sig.		0.469		* 0.04		0.077		0.299		0.257	0.128
**Gender**												
	Sum of Squares		1.066		0.001		0.038		0.002		0.151	0.103
	df		3		1		1		1		1	1
	Mean Square		0.355		0.001		0.038		0.002		0.151	0.103
	F		1.428		0.002		0.153		0.006		0.605	0.413
	Sig.		0.233		0.961		0.695		0.936		0.437	0.521
**Education**												
	Sum of Squares		2.042		0.928		0.663		0.877		0.298	0.486
	df		3		1		1		1		1	1
	Mean Square		0.681		0.928		0.663		0.877		0.298	0.486
	F		1.189		1.621		1.158		1.532		0.519	0.848
	Sig.		0.313		0.203		0.282		0.216		0.471	0.357
**Marital status**												
	Sum of Squares		0.723		1.817		3.39		2.032		1.691	3.624
	df		3		1		1		1		1	1
	Mean Square		0.241		1.817		3.39		2.032		1.691	3.624
	F		0.608		4.592		8.584		5.136		4.273	9.178
	Sig.		0.61		* 0.032		* 0.003		* 0.024		* 0.039	* 0.002
**Geopolitical zone**												
	Sum of Squares		9.83		2.366		2.656		0.182		0.031	0.438
	df		3		1		1		1		1	1
	Mean Square		3.277		2.366		2.656		0.182		0.031	0.438
	F		1.331		0.96		1.078		0.074		0.013	0.178
	Sig.		0.263		0.327		0.299		0.786		0.911	0.674
**Household income**												
	Sum of Squares		6.722		2.487		1.542		1.504		2.11	4.532
	df		3		1		1		1		1	1
	Mean Square		2.241		2.487		1.542		1.504		2.11	4.532
	F		1.688		1.873		1.161		1.132		1.589	3.415
	Sig.		0.167		0.171		0.281		0.287		0.208	0.065
**(c). Awareness of food fraud and recall among respondents.**												
**Variables**	**ANOVA**	**Through Information from the Place Where the Food Was Purchased.**	**Through Information from the Manufacturer (Company or Business) Producing the Food.**	**Through Social Media or News Outlets.**	**Through Communication from a Regulatory Agency, e.g., NAFDAC or Standards Organisation of Nigeria (SON)**	**Through your Research or Investigation.**	**Street Vendors**	**Open Market**	**Supermarket or Store**	**Restaurant**	Warehouse	Manufacturing Facility
**Age**												
	Sum of Squares	0.7	1.171	0.257	0.736	0.2	0	0.824	3.454	2.253	0.8	0.001
	df	1	1	1	1	1	1	1	1	1	1	1
	Mean Square	0.7	1.171	0.257	0.736	0.2	0	0.824	3.454	2.253	0.8	0.001
	F	0.651	1.09	0.239	0.684	0.186	0	0.767	3.219	2.099	0.745	0.001
	Sig.	0.42	0.297	0.625	0.408	0.666	0.992	0.381	0.073	0.148	0.388	0.974
**Gender**												
	Sum of Squares	0.133	1.014	0.331	0.073	0.078	0.474	0.079	0.238	0.367	0.268	0.233
	df	1	1	1	1	1	1	1	1	1	1	1
	Mean Square	0.133	1.014	0.331	0.073	0.078	0.474	0.079	0.238	0.367	0.268	0.233
	F	0.536	4.082	1.33	0.294	0.313	1.906	0.319	0.955	1.477	1.077	0.936
	Sig.	0.464	* 0.043	0.249	0.588	0.576	0.168	0.572	0.329	0.224	0.299	0.333
**Education**												
	Sum of Squares	0.531	0.883	1.243	0.085	0.162	0.469	1.912	1.3	0.655	0.142	0.002
	df	1	1	1	1	1	1	1	1	1	1	1
	Mean Square	0.531	0.883	1.243	0.085	0.162	0.469	1.912	1.3	0.655	0.142	0.002
	F	0.928	1.542	2.171	0.149	0.283	0.819	3.342	2.271	1.143	0.249	0.003
	Sig.	0.336	0.214	0.141	0.7	0.595	0.366	0.068	0.132	0.285	0.618	0.958
**Marital status**												
	Sum of Squares	0.153	0.686	0.207	0.06	0.04	0.225	0.002	0.131	0.135	0.014	0.098
	df	1	1	1	1	1	1	1	1	1	1	1
	Mean Square	0.153	0.686	0.207	0.06	0.04	0.225	0.002	0.131	0.135	0.014	0.098
	F	0.385	1.731	0.522	0.15	0.1	0.567	0.006	0.33	0.341	0.036	0.246
	Sig.	0.535	0.188	0.47	0.699	0.752	0.452	0.939	0.566	0.559	0.85	0.62
**Geopolical zone**												
	Sum of Squares	11.881	19.841	18.597	9.737	13.3	0.038	0.01	5.387	2.07	0.735	5.474
	df	1	1	1	1	1	1	1	1	1	1	1
	Mean Square	11.881	19.841	18.597	9.737	13.3	0.038	0.01	5.387	2.07	0.735	5.474
	F	4.831	8.081	7.568	3.955	5.407	0.015	0.004	2.188	0.84	0.298	2.223
	Sig.	* 0.028	* 0.005	* 0.006	* 0.047	* 0.02	0.901	0.95	0.139	0.359	0.585	0.136
**Household income**												
	Sum of Squares	0.001	0.307	0.003	0.08	0.003	0.117	0.404	0.039	0.011	0.093	0.292
	Df	1	1	1	1	1	1	1	1	1	1	1
	Mean Square	0.001	0.307	0.003	0.08	0.003	0.117	0.404	0.039	0.011	0.093	0.292
	F	0	0.231	0.002	0.06	0.002	0.088	0.304	0.03	0.008	0.07	0.22
	Sig.	0.983	0.631	0.962	0.807	0.963	0.767	0.581	0.863	0.927	0.792	0.639

ANOVA—analysis of variance, * *p* < 0.05 (significant).

**Table 3 foods-13-03270-t003:** Attitudes of Nigerian consumers towards food fraud (n = 2160).

Statement	Responses % (n)				
	Strongly Disagree	Disagree	Neither Agree Nor Disagree	Agree	Strongly Agree
I am generally concerned about counterfeit, adulterated, or mislabelled food items.	5.9 (127)	2.2 (48)	6.3 (136)	26.5 (572)	59.1 (1277)
I am generally concerned about counterfeit, adulterated, or mislabelled food items that are made in Nigeria.	4.2 (90)	3.5 (76)	7.2 (155)	27 (584)	58.1 (1255)
I am generally concerned about counterfeit, adulterated, or mislabelled food items that are imported.	4.2 (91)	4 (87)	12.4 (268)	26.9 (582)	52.4 (1132)
I believe that regulatory agencies, i.e., government agencies like NAFDAC and SON, are the most competent to protect Nigerians from counterfeit, adulterated or mislabelled foods.	8.9 (192)	8.8 (191)	18.1 (392)	28 (605)	36.1 (780)
I believe food producers, manufacturers, and sellers are the most competent to protect Nigerians from counterfeit, adulterated or mislabelled foods.	10.6 (229)	12.9 (279)	20.9 (451)	27.3 (589)	28.3 (612)
I believe it is up to consumers to protect themselves from counterfeit, adulterated, or mislabelled foods when shopping or eating at a restaurant.	9.6 (208)	9.4 (203)	13.4 (289)	18.7 (403)	48.9 (1057)
Food fraud can cause serious health challenges, including death.	9.3 (201)	8.4 (182)	12.8 (277)	20.3 (439)	49.1 (1061)
I believe the government is creating enough awareness about counterfeit, adulterated or mislabelled foods.	24.4 (527)	28.3 (612)	21.3 (460)	16.3 (353)	9.6 (208)
A food product with a NAFDAC number is not counterfeit, adulterated, or mislabeled	21.5 (464)	22.2 (479)	27.5 (593)	17.1 (370)	11.8 (254)
If I suspect a food product is counterfeit, mislabelled, or adulterated, it is easy to report this to regulatory agencies like NAFDAC or SON.	17.6 (380)	25.4 (548)	23.4 (505)	20 (432)	13.7 (295)
During challenging economic times, it is acceptable for food sellers to sell counterfeit or adulterated food items as long as these foods do not cause harm.	52.3 (1129)	16.9 (366)	9.7 (210)	9.8 (212)	11.3 (243)

**Table 4 foods-13-03270-t004:** Pearson’s correlation of respondents’ sociodemographic characteristics versus food fraud awareness and attitudes.

		Gender	Age	Education	Marital Status	Geo-Political Zone	Household Income	Awareness	Attitude
Gender	Pearson Correlation	1	0.068 **	0.025	−0.049 *	−0.161 **	0.062 **	0.027	−0.001
	Sig. (2-tailed)		0.002	0.25	0.022	<0.001	0.004	0.215	0.963
	N	2160	2160	2160	2160	2160	2160	2160	2160
Age	Pearson Correlation	0.068 **	1	0.557 **	0.585 **	0.042	0.445 **	−0.023	−0.026
	Sig. (2-tailed)	0.002		<0.001	<0.001	0.054	<0.001	0.295	0.234
	N	2160	2160	2160	2160	2160	2160	2160	2160
Education	Pearson Correlation	0.025	0.557 **	1	0.334 **	0.036	0.382 **	0.011	0.037
	Sig. (2-tailed)	0.25	<0.001		<0.001	0.091	<0.001	0.596	0.089
	N	2160	2160	2160	2160	2160	2160	2160	2160
Marital status	Pearson Correlation	−0.049 *	0.585 **	0.334 **	1	0.021	0.312 **	−0.023	−0.027
	Sig. (2-tailed)	0.022	<0.001	<0.001		0.324	<0.001	0.276	0.214
	N	2160	2160	2160	2160	2160	2160	2160	2160
Geo-political zone	Pearson Correlation	−0.161 **	0.042	0.036	0.021	1	0.029	−0.037	0.03
	Sig. (2-tailed)	<0.001	0.054	0.091	0.324		0.177	0.086	0.168
	N	2160	2160	2160	2160	2160	2160	2160	2160
Household income	Pearson Correlation	0.062 **	0.445 **	0.382 **	0.312 **	0.029	1	0.024	−0.054 *
	Sig. (2-tailed)	0.004	<0.001	<0.001	<0.001	0.177		0.259	0.012
	N	2160	2160	2160	2160	2160	2160	2160	2160
Awareness	Pearson Correlation	0.027	−0.023	0.011	−0.023	−0.037	0.024	1	0.024
	Sig. (2-tailed)	0.215	0.295	0.596	0.276	0.086	0.259		0.267
	N	2160	2160	2160	2160	2160	2160	2160	2160
Attitude	Pearson Correlation	−0.001	−0.026	0.037	−0.027	0.03	−0.054 *	0.024	1
	Sig. (2-tailed)	0.963	0.234	0.089	0.214	0.168	0.012	0.267	
	N	2160	2160	2160	2160	2160	2160	2160	2160

** Correlation is significant at *p* < 0.01 level (2-tailed), * Correlation is significant at *p* < 0.05 level (2-tailed). The value highlighted in green indicate a statistically significant association between attitudes to food fraud and household income, as determined by Pearson correlation (*p* = 0.012).

## Data Availability

The original contributions presented in the study are included in the article, further inquiries can be directed to the corresponding author.

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
