# Peer review of "Navigating Food Fraud: A Survey of Nigerian Consumer Knowledge and Attitudes"

_foods, 2024, doi:10.3390/foods13203270_

Round 1
Reviewer 1 Report
Comments and Suggestions for Authors
The study explores Nigerian consumer awareness and attitudes toward food fraud, focusing on economically motivated adulteration and counterfeit food products. The research highlights the health, economic, and safety risks posed by food fraud, which includes practices like mislabeling, dilution, and substitution of food products for financial gain.
A survey conducted across Nigeria, involving 2,160 participants, assessed their familiarity with food fraud incidents, frequently adulterated products, and the effectiveness of regulatory bodies such as the National Agency for Food and Drug Administration and Control (NAFDAC). Results revealed that over a third of respondents were unaware of the term "food fraud," although many recognized high-profile local cases like "plastic rice" and counterfeit dairy products. The majority of participants identified street vendors as potential sources of food adulteration.
Potential encouragements are given as under;
1. Around 85% of respondents expressed concern about fraudulent food products, regardless of origin (local or imported).
2. Most respondents relied on point-of-sale information to assess food authenticity, with low awareness of NAFDAC’s recall system (only 50.8% were familiar).
3. A significant portion of the respondents (49.1%) agreed that food fraud can lead to severe health issues, including death.
4. 21.1% of respondents agreed that during economic hardship, it may be acceptable for sellers to offer adulterated products, provided they do not cause harm.
5. Education, income, and geopolitical location significantly influenced awareness levels, with educated and higher-income groups showing more knowledge of food fraud.
Key points needs to be clarify to enhance the quality of paper:
1. The paper touches on consumer attitudes towards food fraud during economic hardships, with some respondents accepting adulteration if it doesn't harm. This point seems significant, but the discussion around it could be expanded. It might be worth exploring in more depth the reasons behind this acceptance and its implications for food safety policies.
2. While the paper uses descriptive statistics and ANOVA to analyze the data, the practical significance of the findings (effect size) is not always clear. There is a need to discuss whether the statistically significant results also have real-world implications, especially in terms of policy-making and consumer protection.
3. The section on consumer awareness of food fraud could be strengthened by providing more in-depth analysis of the geographical distribution of awareness. For example, why certain regions might have lower awareness than others was not deeply explored.
4. The sampling and questionnaire distribution methods, though explained, could benefit from more detail regarding potential sampling biases, particularly with regard to the online dissemination method. There is a possibility that the online survey responses may have skewed towards a more educated or internet-accessible demographic, which might not be representative of the broader Nigerian population.
5. The discussion on the sampling method is missing. Was it stratified random sampling, purposive sampling, convenience sampling, snowball sampling, quota sampling, or another method? The authors should provide justification for their choice.
6. The authors should also explain how authenticity and rigor were maintained in regard to data collection and analysis."
7. Overall, methodology design is poorly constructed and requires substantial development and explanation. There is a lack of convincing justification to support the methodology section.
Comments on the Quality of English LanguageModerate editing of English language required.
Author Response
REVIEWER’S COMMEN AND AUTHOR’S RESPONSE FOR NAVIGATING FOOD FRAUD: A SURVEY OF NIGERIAN CONSUMER KNOWLEDGE AND ATTITUDES
|
S/N |
REVIEW COMMENTS |
AUTHORS’ RESPONSE |
|
|
REVIEWER 1 |
|
|
1. |
The paper touches on consumer attitudes towards food fraud during economic hardships, with some respondents accepting adulteration if it doesn't harm. This point seems significant, but the discussion around it could be expanded. It might be worth exploring in more depth the reasons behind this acceptance and its implications for food safety policies. |
Thank you for your valuable feedback. The discussion on this point has been expanded with reasons behind this acceptance and its implications for food safety policies. |
|
2. |
While the paper uses descriptive statistics and ANOVA to analyze the data, the practical significance of the findings (effect size) is not always clear. There is a need to discuss whether the statistically significant results also have real-world implications, especially in terms of policy-making and consumer protection. |
Thank you for your valuable feedback. We acknowledge the lack of effect size to underscore the practical significance of the findings as a limitation in the study. At the same time, we have stressed how the findings in the study will address policy in this regard. |
|
3. |
The section on consumer awareness of food fraud could be strengthened by providing more in-depth analysis of the geographical distribution of awareness. For example, why certain regions might have lower awareness than others was not deeply explored. |
Thank you for your valuable feedback. Further explanation has been provided on why certain regions had lower awareness than others. |
|
4. |
The sampling and questionnaire distribution methods, though explained, could benefit from more detail regarding potential sampling biases, particularly with regard to the online dissemination method. There is a possibility that the online survey responses may have skewed towards a more educated or internet-accessible demographic, which might not be representative of the broader Nigerian population. |
We appreciate the reviewer’s insightful comments regarding potential sampling biases in the online survey distribution method. We acknowledge that the online dissemination of the questionnaire could have introduced a bias towards participants who have internet access and are potentially more educated. To address this, we have revised the manuscript to provide more detail about the steps taken to mitigate this bias. |
|
5. |
The discussion on the sampling method is missing. Was it stratified random sampling, purposive sampling, convenience sampling, snowball sampling, quota sampling, or another method? The authors should provide justification for their choice. |
Thank you for your valuable feedback. A justification on the sampling method has been provided along with the justification for the used sampling methods. |
|
6. |
The authors should also explain how authenticity and rigor were maintained in regard to data collection and analysis." |
Thank you for your valuable feedback. An explanation on how authenticity and rigor were maintained in regard to data collection and analysis has been made. |
|
7. |
Overall, methodology design is poorly constructed and requires substantial development and explanation. There is a lack of convincing justification to support the methodology section. |
A justification to support the methodology has been included in the discussion. |
Reviewer 2 Report
Comments and Suggestions for Authors
The authors investigate by using a survey the knowledge, awareness and attitude of Nigerians regarding food fraud. Their study of 2,160 Nigerian consumers finds that while over a third were unfamiliar with the term "food fraud," many were aware of high-profile cases like "plastic rice" and counterfeit dairy products, with concerns particularly about food from street vendors. The study reveals a need for better consumer awareness and stronger regulatory measures to combat widespread adulteration, especially in products like alcoholic drinks, dairy, honey, rice, and tomato paste.
In my view, the study is convincing, and is informative for readers. The paper is well structures and written. The conclusions are derived from the results.
I have only two remarks, maybe it make sense to refer the a recent food fraud executed by Nestle. See e.g.
https://www.foodwatch.org/en/new-revelations-in-the-mineral-water-scandal-nestle-has-apparently-been-using-illegal-filtering-methods-for-decades
Secondly, I would like the authors to consider the argument that not only are consumers victims of cheaters, but so are honest companies. Honest competitors also suffer from the actions of cheaters, and the economic consequences can be far-reaching. The argument is as follows: Cheaters obviously try to increase their sales through food fraud, which typically reduces their costs. As a result, honest competitors lose sales and market share. In extreme cases, honest competitors may even go bankrupt. If the fraud persists, the only way for an honest competitor to catch up with the cheater might be to engage in cheating as well. Ultimately, if all companies begin to cheat consumers, it is unlikely that any legal consequences will follow, as policymakers and judges may be reluctant to hold an entire industry accountable.
Even if fraud is detected, honest competitors may still be harmed if consumers substitute the affected kind of product with another. This would only occur if the product has close substitutes and if consumers lose trust in the integrity of the entire industry.
Author Response
|
1. |
maybe it make sense to refer the a recent food fraud executed by Nestle. See e.g.
https://www.foodwatch.org/en/new-revelations-in-the-mineral-water-scandal-nestle-has-apparently-been-using-illegal-filtering-methods-for-decades |
Thank you for the suggestion. We have incorporated a reference to the recent food fraud case involving Nestlé, as outlined in the Foodwatch report. This has been added to the revised manuscript to further illustrate the significance of real-world examples in highlighting the need for stronger regulatory measures. |
|
2. |
Secondly, I would like the authors to consider the argument that not only are consumers victims of cheaters, but so are honest companies. Honest competitors also suffer from the actions of cheaters, and the economic consequences can be far-reaching. The argument is as follows: Cheaters obviously try to increase their sales through food fraud, which typically reduces their costs. As a result, honest competitors lose sales and market share. In extreme cases, honest competitors may even go bankrupt. If the fraud persists, the only way for an honest competitor to catch up with the cheater might be to engage in cheating as well. Ultimately, if all companies begin to cheat consumers, it is unlikely that any legal consequences will follow, as policymakers and judges may be reluctant to hold an entire industry accountable.
Even if fraud is detected, honest competitors may still be harmed if consumers substitute the affected kind of product with another. This would only occur if the product has close substitutes and if consumers lose trust in the integrity of the entire industry. |
We appreciate the reviewer’s insightful comment regarding the impact of food fraud on honest competitors and the broader market dynamics. In response, we have expanded our discussion to include this argument. We now emphasize how food fraud not only harms consumers but also negatively impacts honest companies through lost sales, reduced market share, and the risk of bankruptcy. |
Round 2
Reviewer 1 Report
Comments and Suggestions for Authors
The introduction section should have a research question or real problem. Please add one real example regarding the results of this study.
Comments on the Quality of English LanguageThe introduction section should have a research question or real problem. Please add one real example regarding the results of this study.